# BGF-YOLOv10: Small Object Detection Algorithm from Unmanned Aerial Vehicle Perspective Based on Improved YOLOv10

**DOI:** 10.3390/s24216911

**Published:** 2024-10-28

**Authors:** Junhui Mei, Wenqiu Zhu

**Affiliations:** School of Computer Science, Hunan University of Technology, Zhuzhou 412007, China; m22085400006@stu.hut.edu.cn

**Keywords:** UAV, object detection, BGF-YOLOv10, VisDrone-DET2019, UAVDT

## Abstract

With the rapid development of deep learning, unmanned aerial vehicles (UAVs) have acquired intelligent perception capabilities, demonstrating efficient data collection across various fields. In UAV perspective scenarios, captured images often contain small and unevenly distributed objects, and are typically high-resolution. This makes object detection in UAV imagery more challenging compared to conventional detection tasks. To address this issue, we propose a lightweight object detection algorithm, BGF-YOLOv10, specifically designed for small object detection, based on an improved version of YOLOv10n. First, we introduce a novel YOLOv10 architecture tailored for small objects, incorporating BoTNet, variants of C2f and C3 in the backbone, along with an additional small object detection head, to enhance detection performance for small objects. Second, we embed GhostConv into both the backbone and head, effectively reducing the number of parameters by nearly half. Finally, we insert a Patch Expanding Layer module in the neck to restore the feature spatial resolution. Experimental results on the VisDrone-DET2019 and UAVDT datasets demonstrate that our method significantly improves detection accuracy compared to YOLO series networks. Moreover, when compared to other state-of-the-art networks, our approach achieves a substantial reduction in the number of parameters.

## 1. Introduction

In recent years, the rapid development of deep neural networks, exemplified by object detection in computer vision, has empowered unmanned aerial vehicles (UAVs) with the ability to autonomously perceive, make decisions, and analyze their surroundings. This capability has enabled UAVs to play an increasingly important role in real-world applications. By utilizing object detection technology, UAVs can accurately identify target objects of interest, thereby enhancing the efficiency and effectiveness of data collection. As a result, UAVs hold great potential for widespread use in various fields such as traffic monitoring, disaster relief, and agricultural crop inspection.

However, in UAV applications, the resolution of objects viewed from the UAV’s perspective is often around 32 × 32 pixels. These objects, typically referred to as “small targets”, pose a challenge when constructing object detectors for UAV platforms. These challenges primarily manifest in two aspects. On one hand, to achieve high accuracy in detecting small objects within high-resolution UAV images, some studies have developed complex models; however, these models incur substantial computational costs. On the other hand, small objects constitute a significant proportion of aerial images while possessing limited visual features, making it difficult for detectors to accurately identify them.

To address the balance between accuracy and efficiency, many researchers have proposed solutions, with most focusing on model compression to reduce the number of parameters in the network. For instance, SlimYOLOv3 [1] employs pruning techniques to decrease model complexity, significantly lowering the energy consumption required by the network. UAV-Net [2] adapts model pruning and improves the detection layers. While these approaches facilitate deployment on UAV platforms, they often struggle with convergence during training and exhibit poor robustness. Recently, ShuffleDet [3] combined channel shuffling with grouped convolutions to increase network width for multi-scale detection, effectively mitigating computational redundancy by avoiding extensive 1 × 1 convolution operations. Mobiledets [4] constructs a search space by designing regular convolution blocks, achieving a better balance between latency and accuracy across various mobile devices. QueryDet [5] implements a query mechanism to accelerate the inference speed of multi-scale object detection while leveraging high-resolution feature maps for detailed information, thus avoiding ineffective processing of background areas. Although these studies have led to improved performance in specific scenarios, significant accuracy gaps remain compared to state-of-the-art detectors.

Recently, YOLOv10 [6] has achieved an efficient architecture through the design of a lightweight classification head, spatial-channel decoupling downsampling, and ranking-guided blocks. As shown in Figure 1, we propose an improved algorithm based on YOLOv10 [6], named BGF-YOLOv10 (BoTNet [7], GhostConv [8], Patch Expanding Layer), specifically tailored for small object detection in UAV perspectives. as shown in Figure 1. First, we add an additional detection head to YOLOv10 and insert BoTNet’s multi-head attention module, C2f_faster, and C3_faster into the backbone and neck, respectively, to enhance the detection of small objects. To reduce model parameters, we replace standard convolutions with GhostConv in both the backbone and neck. Finally, since small objects contain limited feature information, we designed a Patch Expanding Layer, an upsampling module that effectively extracts contextual information around small objects. Experimental results show that our BGF-YOLOv10 significantly outperforms YOLOv10 on the VisDrone2019 and UVDAT datasets. Our contributions are as follows:
We propose a novel network architecture for small object detection and validate its effectiveness through experiments based on BGF-YOLOv10. By incorporating the multi-head attention module from BoTNet [7] into the backbone to enhance the model’s capability to capture information, and by introducing the improved C2f and C3 modules into the neck, we further improve detection speed. We integrate GhostConv [8] modules into both the backbone and the neck, which significantly reduces network parameters without sacrificing detection accuracy.We designed a Patch merging layer, an upsampling module that adds feature information during the upsampling process, enhancing small object detection by leveraging contextual information.

## 2. Related Work

### 2.1. YOLO Network

Among the many real-time object detection algorithms, the YOLO [9] series (V1 to V10) has developed rapidly, significantly influencing various fields of computer vision research. YOLOv5 followed the design philosophy of YOLOv3 [10], integrating CSPNet [11] and CSPPAN from YOLOv4 while optimizing the speed and accuracy of the CSP fusion layer and activation functions. YOLOv7 replaced CSPNet [11] with ELAN and introduced E-ELAN for designing larger models. YOLOv8 refined YOLOv7’s ELAN architecture by adding additional residual connections, with an encoder similar to YOLOv6. YOLOv10 [6] built upon the YOLOv6 architecture and significantly improved global feature extraction by incorporating transformer-based modules. The one-to-many and one-to-one matching design allowed the model to perform end-to-end object detection without the need for additional NMS post-processing.

SCA-YOLO [12] designed a hybrid attention module based on a coordinate attention mechanism to enhance feature extraction for small objects. GCL-YOLO [13] proposed a specialized prediction head for small objects in drone images, particularly targeting small objects in complex environments. SOAR [14] introduced a lightweight YOLOv9 architecture that utilizes the SAHI framework, leveraging Programmable Gradient Information (PGI) to mitigate the substanti information loss typically encountered during sequential feature extraction processes. The paper employs the Vision Mamba model, which incorporates positional embeddings to facilitate precise location-aware visual understanding. Additionally, it integrates a novel bidirectional State Space Model (SSM) for efficient visual context modeling. This SSM cleverly exploits the linear complexity of CNNs and the global receptive field of transformers, making it particularly effective for remote sensing image classification. VIT-YOLO [15] integrates a hybrid detector that combines transformers and YOLO, enhancing feature extraction and multi-scale feature fusion through the use of MHSA-Darknet and BiFPN.

YOLOv10 [6] also optimized label assignment through dual-label allocation and stop-gradient operations, ensuring precise label matching. Furthermore, the ranking-guided block design enabled the model to select convolution types based on different stages, enhancing computational efficiency. The partial self-attention mechanism combining CSPNet [11] and transformers further enhanced the model’s expressiveness and robustness. These innovations have led YOLOv10 [6] to achieve higher accuracy, efficiency, and flexibility in object detection tasks.

### 2.2. Small Object Detection

Small objects in images often exhibit varying scales, which introduces different levels of detection difficulty for a single detector. Techniques such as image pyramids [16] and sliding window schemes are commonly employed to address scale variations [17,18]. Some studies choose scale-specific detectors to tackle this challenge. For example, SSH [19] integrates scale-specific detectors trained for certain scale ranges to handle faces of extreme sizes. TridentNet [20] constructs a parallel multi-branch architecture, where each branch is optimized for the receptive field best suited to a particular scale. QueryDet [5] introduces a cascade query strategy, which avoids computations on low-level features and enables efficient small object detection at high resolution. Reference [21] proposes a feedback-driven training paradigm to dynamically guide data preparation, further balancing the training loss for small objects.

In small object scenarios, small objects in high-resolution images are often unevenly distributed, leading to redundant computations when dividing the detection process. To address this issue, CLusDet [22] leverages semantic and spatial parameters between objects to integrate detection, while CRENet [23] designs a clustering algorithm that adaptively searches for clustered regions. F&S [24] introduces the focus-and-detect framework, where the Focusing Network identifies candidate regions and then enhances the resolution of those regions, enabling accurate detection of small objects. Compared to traditional sliding window mechanisms, the focus-and-detect approach offers significant advantages through adaptive cropping and flexible magnification operations. This allows smaller objects to be processed at higher resolutions, while larger objects can be detected at lower resolutions, significantly saving memory and reducing background interference during inference.

From the perspective of information theory, the more feature types a model can capture, the higher its detection accuracy tends to be. Some studies leverage contextual information to enhance small object detection. FS-SSD [25] refines the detection of low-confidence objects by utilizing the intra-class and inter-class instance distances. SINet [26] introduces a context-aware pooling layer to preserve contextual information, while R2-CNN [27] employs a global attention mechanism to suppress redundant computations and effectively detect small objects in large-scale remote sensing images. From an information-theoretic standpoint, considering a broader range of feature types generally leads to higher detection precision. Based on this understanding, researchers have extensively explored context-driven methods aimed at generating more discriminative features, particularly for small objects with limited cues, thereby improving detection accuracy. However, current context modeling methods—whether global or local—face challenges in determining which regions should be encoded as context. In other words, these mechanisms often rely on heuristic and empirical approaches to define contextual regions, which may not guarantee that the constructed feature representations are sufficiently interpretable for detection tasks.

From the perspective of multi-scale and global weighting, CFP [28] enhances object detection performance by globally modulating feature pyramids through a focused feature pyramid network, effectively integrating global and local information. SSRDet [29] introduces a scale enhancement module and a scale selection module to extend the feature pyramid network. ISOD [30] combines extended-scale feature pyramid networks and high-resolution pyramids to improve small object detection capabilities. Mini-YOLOv4 [31] incorporates a hierarchical feature pyramid to facilitate the extraction of fine-grained features. G2Grad-CAMRL [32] adds global average pooling layers to obtain feature weight vectors and overlays the weighted vector with the output class activation map. CEASC [33] employs context-enhanced group normalization and an adaptive multi-layer masking strategy to effectively improve detection.

## 3. Method

The original YOLOv10 replaced the dual-head mechanism with a one-to-many and one-to-one matching strategy, enabling end-to-end object detection. Building on the default model, we propose three key improvements: (1) a novel YOLOv10 architecture specifically designed for small object detection scenarios; (2) the integration of the GhostConv module into both the backbone and neck; and (3) the introduction of a Patch Expanding Layer, an upsampling module that enhances feature information during upsampling and leverages contextual information to improve small object detection.

### 3.1. Novel YOLOv10 Architecture

To enhance baseline detection performance while reducing the number of parameters and introducing minimal additional latency, we replaced the original PSA module in the YOLOv10 backbone with the BoTNet [7] layer. The core idea of the multi-head self-attention (MHSA) mechanism is to compute the relationships between each position in the input sequence and all other positions simultaneously using multiple attention heads, thereby capturing different feature subspaces. Convolutional models excel in parameter sharing and efficient local information aggregation, but acquiring global information requires stacking numerous convolutional layers. In contrast, the self-attention mechanism inherently captures global information, reducing the model’s depth. BoTNet [7] replaces the last three spatial (3 × 3) convolutions in ResNet with multi-head self-attention (MHSA). The memory requirement of the self-attention mechanism scales quadratically in the spatial dimension, leading to high memory and computational demands when processing large input images. By first using convolutional layers to extract low-resolution features, ResNet reduces the memory footprint before feeding them into the self-attention mechanism, as shown in Figure 2.

To maintain inference speed, we drew inspiration from FasterNet and replaced the Bottleneck in C2F with FasterBlock, achieving higher floating-point operations (FLOPs). FasterNet’s PConv (Partial Convolution) plays a crucial role in reducing computational redundancy and memory usage. In facilitating continuous memory access, PConv uses the number of channels as a representative of the entire feature map for computation. The FLOPs of the convolutional layer are expressed as shown in Equation (1):(1)$$FLOPs_{C\mathrm{onv}}=h\times w\times k^{2}\times c^{2}$$
where h×w denote the height and width of the input image, k represents the kernel size, and c is the number of channels. If the partial convolution ratio is r=1/4, it is equivalent to 1/16 of a standard convolution. Although Bottleneck uses 1 × 1 convolutions to reduce the number of channels and computation, its computational cost is still higher compared to Faster_Block. FasterBlock, through PConv, performs convolution operations on partial channels, significantly reducing computational demands and memory access, making it particularly well-suited for resource-constrained hardware. The structural design of Faster_Block, incorporating PConv and two PWConv layers, enables feature extraction and channel fusion, and employs shortcut connections to retain input features, further minimizing redundant computations. Inspired by these advantages, we incorporated FasterBlock into the YOLOv10 network, replacing Bottleneck with FasterBlock in both C2f and C3, as illustrated in Figure 3.

Leveraging the lightweight advantages of BoTNet [7] and C2f_faster, as illustrated in the figure, we modified the YOLOv10 architecture. The primary role of the backbone is to extract relevant features from the input data, which constitutes the main part of the network. Our objective focuses on improving accuracy while maintaining lightweight characteristics. Although PSA offers robust feature enhancement, its computational efficiency does not match that of BoTNet [7]. Specifically, PSA imposes significant computational overhead when processing high-dimensional features. Therefore, we opted to replace PSA with the entire BoTNet [7] network, which provides stronger feature fusion capabilities and a more flexible design. BoTNet [7] combines Bottleneck Transformer and multi-head self-attention (MHSA), enabling more effective capture of complex contextual information and enhancing feature representation through its CSP structure. After replacing the PCA layer with the BoTNet layer and training from scratch on the VisDrone2019 dataset, we observed a significant improvement in accuracy, although the network parameters increased slightly. This is discussed in detail in Section 4.4.2. Therefore, we chose to replace PSA with BoTNet [7], which provides stronger feature fusion capabilities and a more flexible network design. BoTNet [7] integrates the Bottleneck Transformer and multi-head self-attention (MHSA), allowing it to more effectively capture complex contextual information and enhance feature representation through its CSP structure. Additionally, we replaced most of the C2f and C3 blocks with the FastertBlock-integrated C2F_faster and C3_faster in both the backbone and neck parts, resulting in a substantial speed improvement during detection. To ensure accuracy, we added an additional detection head specifically for small objects to the YOLOv10 base model. Details of this enhancement will be discussed in Section 3.4.

### 3.2. GhostConv Model

A standard convolutional module typically consists of a batch normalization layer, a conventional convolution layer, and an activation function. This configuration often results in a large number of similar feature maps, leading to high computational costs and memory consumption. GhostNet [8] suggests that the redundant features within feature maps can still help neural networks achieve good detection accuracy. To reduce these demands, Ghost Convolution (GhostConv) adopts a two-step strategy as illustrated in Figure 4. First, it applies standard convolution with small kernels to generate a reduced number of feature maps. Then, depthwise convolution is employed to generate additional feature maps that were not created in the first step. Finally, the feature maps from both steps are fused to produce a final feature map similar to that of a standard convolution layer but with significantly lower computational and parameter costs. Compared to the standard convolution module, GhostConv uses half of the feature channels for 5 × 5 depthwise convolution, which significantly expands the receptive field in a single step, and then restores the two feature channels to the original features. Therefore, we incorporated the GhostConv module into the YOLOv10 [6] network to reduce both the number of parameters and the computational complexity. Additionally, GhostConv leverages lower-cost computations to obtain these redundant feature maps. In the backbone, traditional convolution is employed to extract a portion of the true feature maps, while in the neck, channel compression is applied to reduce the impact of noise.

### 3.3. Patch Merging Layer

In target images captured by drones, small objects are unevenly distributed across the entire image and occupy only a small number of pixels. The features of small objects are often concentrated in neighboring pixels. We leverage this local contextual information to enhance the perception of small objects. The module divides the input feature map into four patches. Assuming the input feature has a dimension of W×H×8C, the first step in the linear layer increases the input feature dimension from 8C to 16C. Here, W×H denote the height and width of the input image, and C is the number of channels.
(2)$$Input(W\times H\times 8C)\overset{L\mathrm{inear}L\mathrm{ayer}}{\to }W\times H\times 16C$$

Next, a rearrangement operation is applied to expand the spatial resolution of the input by a factor of 2, while reducing the feature dimension to 1/4 of its original size (i.e., from 16C to 4C). This process performs upsampling while simultaneously reducing the number of feature channels, thereby maintaining a balance of feature information.
(3)W×H×16C→Rearrange2W×2H×4C

Additionally, small objects are more susceptible to noise and background interference. By merging patches, local information from multiple regions is integrated, enhancing the model’s robustness to noise and allowing it to focus more on the target itself rather than background noise. The Patch Merging Layer aids small object detection by enhancing local contextual information, retaining features after spatial downsampling, and reducing the computational load to allow for deeper networks. Although spatial resolution decreases, the effective combination of contextual information ensures that small object features are preserved, leading to better recognition and distinction during feature extraction and classification. We replaced the upsampling layer with a Patch Merging Layer to enhance contextual information. The specific effects of this modification are illustrated in the heatmaps presented in Section 4.4.2, showing notable improvements in performance. The Patch Merging Layer is innovatively integrated into the neck of YOLOv10 to enhance receptive fields and improve the detection of small objects. This module allows for more efficient feature representation, outperforming the conventional feature pyramid. 

The GhostConv module reduces computational costs by generating a minimal set of feature maps while still capturing essential features. The Patch Expanding Layer complements GhostConv by integrating local contextual information from multiple regions. This module enables the model to efficiently preserve and leverage contextual details, which is particularly beneficial for small object detection, as small objects are often affected by noise and background interference. By combining these two modules, we achieve a broader receptive field without incurring significant computational overhead. The efficient feature extraction of GhostConv, coupled with the Patch Expanding Layer’s enhancement of local context, ensures that small object features are more prominent during the detection process.

### 3.4. Detection Head for Small Objects

The classification head of YOLOv10 adopts a lightweight architecture, consisting of two Depthwise Convolutions with a kernel size of 3×3, followed by a 1×1 convolution. While the original YOLOv10 architecture includes three detection heads with resolutions of 80×80, 40×40, and 20×20, which significantly enhances detection capabilities in various scenarios, its performance in detecting small objects remains suboptimal. Small objects occupy fewer pixels and are prone to being overlooked. Convolutional layers play a critical role in feature extraction, but as the network depth increases, the resolution of the feature maps decreases, making it more challenging for the model to capture fine details of small objects. To address this issue, we introduced an additional detection head dedicated to small object detection, with a resolution of 160×160, to improve accuracy in identifying small objects.

Each prediction head in YOLO takes the fused features extracted by the backbone and neck as input. The final input consists of a vector containing the regressed bounding box, bounding box confidence, and object class predictions. Before generating the final bounding box, anchor boxes are generated using the k-means algorithm based on the dataset, with three distinct scales defined to accommodate the detection of small, medium, and large objects. Similarly, we apply k-means clustering to generate anchor boxes for the additional prediction head, specifically designed for improved detection of small objects.

## 4. Experimentation and Results

In this section, we quantitatively evaluate our BGF-YOLOv10 framework to assess its performance and versatility in various complex scenarios. We compare it with the current YOLO networks and lightweight drone object detection models on the VisDrone2019 and UAVDT datasets. The experimental results demonstrate that our proposed BGF-YOLOv10 significantly improves detection accuracy, showcasing its effectiveness across diverse detection tasks.

### 4.1. Implementation Details

In this experiment, the CPU used was a 13th Gen Intel Core i7-13700KF, and the GPU was a single NVIDIA 4090 with 24 GB of memory. The algorithm was implemented using PyTorch 2.0.1 and accelerated with CUDA 11.8. Training was conducted on a single card, with each network trained for 100 epochs. The optimizer used was Adam, with an initial learning rate set to 0.01 and a final learning rate of 0.01. The input images were resized to 640 × 640 pixels, the momentum was set to 0.937, the batch size to 16, and the weight decay to 0.0005. To expedite the experimentation process and ensure consistent results, we applied a range of data augmentation methods tailored to different model sizes. For fairness, all models were trained, validated, and tested from scratch on the same hardware platform and software environment.

### 4.2. Dateset

The VisDrone2019-DET dataset was collected by the AISKYEYE team from the Machine Learning and Data Mining Laboratory of Tianjin University, China. This benchmark dataset consists of 10,209 static images. Figure 5a visualizes the class distribution in VisDrone2019, which includes 10 categories (pedestrian, people, bicycle, car, van, truck, tricycle, awning-tricycle, bus, and motor). There is a significant imbalance in the number of labels for different object classes, leading to uneven label distribution across the dataset. The label positions in Figure 5b show that the dataset contains a large number of small and tiny objects. The UAVDT dataset consists of 23,258 training images and 15,069 test images with a resolution of 1024 × 540. For our experiments, we only used images containing the “car” class.

### 4.3. Evaluation Criteria

In object detection tasks, model performance is typically evaluated using the following metrics: Precision, Recall, and Mean Average Precision (mAP). These metrics are commonly used to measure the effectiveness of detection. Additionally, model complexity is assessed using the number of parameters and Floating Point Operations (FLOPs). True Positives (TPs) refer to instances where both the actual label and the prediction are positive. False Negatives (FNs) occur when the label is positive but the prediction is incorrectly negative. False Positives (FPs) occur when the actual label is negative but the prediction is positive. True Negatives (TNs) represent cases where both the label and the prediction are negative. The formulas for calculating Precision and Recall are as follows:(4)$$pre=\frac{TP}{TP+FP}$$
(5)$$Recall=\frac{TP}{TP+FN}$$

In object detection, the accuracy of a model’s predictions is evaluated by comparing the predicted results with the ground truth using the Intersection over Union (IoU) metric. IoU measures the degree of overlap between the predicted bounding box and the actual ground truth bounding box. It is calculated as the ratio of the area of the intersection to the area of the union between the predicted and ground truth boxes. The IoU value ranges from 0 to 1, where 1 indicates perfect overlap and 0 indicates no overlap. The formula for calculating IoU is as follows:(6)$$IOU=\frac{A\cap B}{A\cup B}$$

In this context, A represents the area of the predicted bounding box, and B represents the area of the ground truth bounding box. Generally, using a higher IoU threshold means stricter accuracy requirements for the prediction results, as the predicted box needs to have a higher overlap with the ground truth box to be considered correct.

In object detection tasks, the overall performance of the model is typically measured by the Average Precision (AP). AP is a stringent evaluation metric used to assess the prediction accuracy of all object categories in the dataset at a specific Intersection over Union (IoU) threshold. The calculation of AP involves precision and recall, with a precision-recall (P-R) curve plotted by measuring the relationship between the two. The x-axis represents recall, while the y-axis represents precision. The AP score is derived from the area under the P-R curve, and it is computed as the weighted average of precision scores at different thresholds. Mean Average Precision (mAP) is the mean of the AP scores across all categories. Specifically, mAP@0.5 refers to the mAP when the IoU threshold is 0.5, while mAP@[0.5:0.95] represents the average mAP over IoU thresholds ranging from 0.5 to 0.95.
(7)$$mAP=\frac{1}{n}\sum_{i=1}^{n}\left(\int_{0}^{1}Precision(Recall)d(Recall\right){)}_{i}$$

### 4.4. Experimental Analysis

#### 4.4.1. Comparative Experiments

Due to the small and dispersed nature of small objects within an image and the fact that traditional YOLO models typically use Non-Maximum Suppression (NMS) for post-processing, there is a certain delay during inference, which is not ideal for deploying models on mobile devices like drones. Therefore, we selected YOLOv10n, introduced by the research team from Tsinghua University, as our baseline within the YOLO series. After implementing our improvements, our model, BGF-YOLOv10, was compared with YOLOv5n and YOLOv8n, which perform well in small object scenarios within the YOLO series, as well as with the original YOLOv10. Our experiments were conducted on the VisDrone2019 dataset, as shown in Table 1.

We have bolded and underlined the best-performing results for each metric. As shown in the table, our model has approximately 0.3M to 1.2M fewer parameters compared to YOLOv10n and YOLOv8n, with a slight reduction in computational cost as well. In terms of mAP50, our model exhibits an overall accuracy improvement of 1 to 3 points over YOLOv10 and approximately 2 points over YOLOv8. Specifically, our model performs well in categories such as pedestrian, car, and truck, but shows some deficiencies in bicycle and tricycle detection. Overall, compared to other YOLO models, our proposed YOLOv10 model for small objects demonstrates relatively accurate detection. BGF-YOLOv10 offers improved overall mAP50 and detection performance across multiple categories while maintaining lower computational complexity (GFLOPs) and parameter count. Notably, it shows significant improvements in key categories such as bicycle, car, truck, and tricycle. This indicates that the model can provide a better balance of efficiency and accuracy in practical applications.

To further validate the effectiveness of our model, we conducted additional comparisons with other methods. Our approach achieves the best performance while utilizing the fewest parameters. It is worth noting that there may be some differences in object scale distribution between the validation and test sets. The experimental results on the VisDrone2019 dataset are shown in Table 2, where our method, BGF-YOLOv10, outperforms all other approaches by achieving the highest mAP with the smallest number of parameters. Overall, BGF-YOLOv10 demonstrates superior detection performance while maintaining real-time efficiency compared to other state-of-the-art models under extreme conditions.

Additionally, we conducted experiments on the UAVDT dataset, with results presented in Table 3. We compare our method against existing approaches across three metrics: mAP50, mAP, and FPS, achieving values of 42.0, 27.2, and 44, respectively. Considering the potential differences in scale variations of small objects between UAVDT and VisDrone2019, our model demonstrates notable improvements on UAVDT. Overall, our method exhibits significant performance gains compared to the baseline. We have bolded and underlined the best-performing results for each metric.

#### 4.4.2. Ablation Study

To demonstrate the effectiveness of the proposed BGF-YOLOv10 network, we conducted an ablation study based on the baseline network. The experiments were performed in a stacked manner, progressively adding modules to the baseline network to compare the effectiveness of each module, as shown in Table 4. The design of the ablation study is as follows. First, we tested the performance of each module incrementally within the baseline network, then evaluated the effects of our proposed new framework with only BoTNet [7] and Faster_Block. This was followed by testing the framework specifically designed for small-object detection, which includes BoTNet [7] + Faster_Block + Detection Head. Second, we tested the replacement of all convolutional layers in the backbone and neck with GhostConv modules. Third, we introduced the proposed PatchExpand module between the backbone and neck. Finally, we incorporated all modules to observe the overall effect. In the Table 4, a √ indicates that we have added this module to the experiment

From the results in Table 4, we observe the following:
(1)Adding BoTNet [7] to the backbone improves precision but also increases the parameter count and computational complexity compared to the baseline. However, incorporating Faster_Block effectively addresses the issue of increased parameter count.(2)Replacing all convolutional layers with GhostConv modules reduces both parameter count and computational load. Nonetheless, the model’s precision decreases due to a reduction in feature channels.(3)Adding an additional detection head significantly enhances the detection of small objects, achieving the best performance in recall. The mAP50 is improved by approximately 1–2 points compared to the baseline.(4)Introducing the proposed PatchExpand module between the backbone and neck shows improvements in both precision and parameter count. This validates the usefulness of contextual information around small objects for their detection, as discussed in Section 3.3.(5)Testing the new BGF-YOLOv10 framework, specifically designed for small-object detection, demonstrates a reduction of 0.3 million parameters and an approximate 2-point improvement in mAP over the baseline.

As illustrated in Figure 6, we compared the feature heatmaps and prediction results extracted using PatchExpand and GhostConv with those obtained using upsample and conv layers in the baseline.

We compared the feature maps extracted by GhostConv and Conv at the backbone location. (A)–(C) depict the heat maps generated by the GhostConv module in the backbone of our proposed BGF-YOLOv10, the contextual information provided by the PatchExpand module, and the final prediction of BGF-YOLOv10, respectively. (D)–(F) show the corresponding modules from the original YOLOv10 prior to replacement: (D) presents the heatmap from the standard Conv module in the backbone, (E) shows the contextual heatmap from the original upsampling layer in the neck, and (F) demonstrates the final prediction map of YOLOv10. From (A) and (D), it is observed that the features extracted by GhostConv are clearer compared to those obtained with Conv. As shown in (B), the feature maps extracted using PatchExpand, as discussed in Section 3.3, demonstrate that enhanced local context information helps the model focus more on the target itself rather than the background. The comparative detection results between the baseline and our approach are shown in (C) and (F) of Figure 6.

As shown in Figure 7, the confusion matrix illustrates the model’s classification performance across different categories. The main diagonal shows the number of correctly classified instances, with the “car” category achieving 9656 correct classifications and the “pedestrian” category achieving 2368, indicating strong performance. Notable misclassifications include “car” being incorrectly classified as “van” 716 times and “pedestrian” being misclassified as “people” 219 times, indicating confusion between these categories. Background classes also exhibit some misclassifications, such as “pedestrian”, “bicycle”, and “people”, highlighting challenges in distinguishing between objects and backgrounds. Rare categories like “awning-tricycle” and “bus” show more significant misclassifications, reflecting issues with long-tail data imbalance.

As shown in Figure 8, the overall average precision (mAP) at an IoU threshold of 0.5 for all classes is 0.317, which represents an improvement of 0.021 over the baseline. Although the PR curves for most categories show a decrease from the first to the second plot, particularly for categories like “pedestrian” and “motor”, where both precision and recall have improved, the overall curves for each category are closer to the lower left corner, indicating a decrease in precision. The performance of the “bus” category shows a slight reduction, but the curve in plot (b) demonstrates an improvement over plot (a), with a slight increase in mAP.

## 5. Conclusions

In this paper, we present YOLOv10, a framework specifically designed for small object detection. To enable efficient deployment on resource-constrained devices such as drones, we leverage the lightweight advantages of GhostConv and Faster_Block modules, allowing the model to improve accuracy while maintaining a lightweight parameter count. Additionally, to enhance small object detection by utilizing contextual information, we designed an upsampling module. The experimental results demonstrate that our model achieves strong performance on both the VisDrone2019 and UDVDT datasets, with relatively low GLOPs. However, our experiments also reveal issues such as background confusion and class imbalance in the dataset. In future work, we plan to address these challenges to further improve our model’s performance.

## Figures and Tables

**Figure 1 sensors-24-06911-f001:**
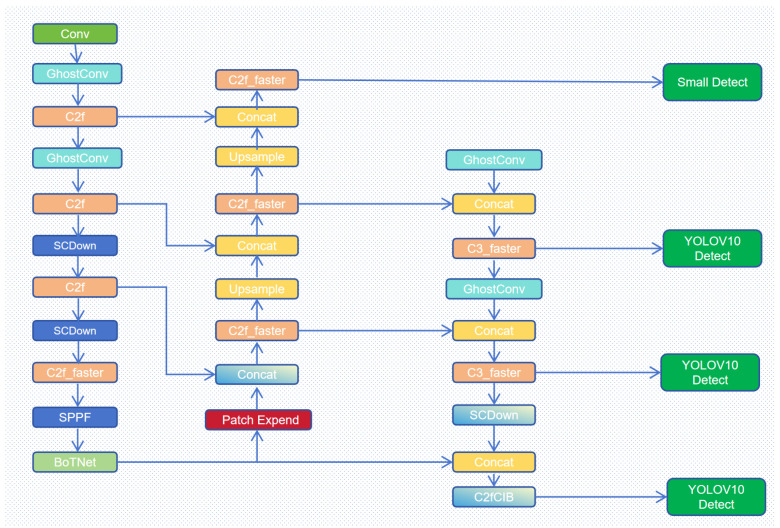
BGF-YOLOv10 framework.

**Figure 2 sensors-24-06911-f002:**
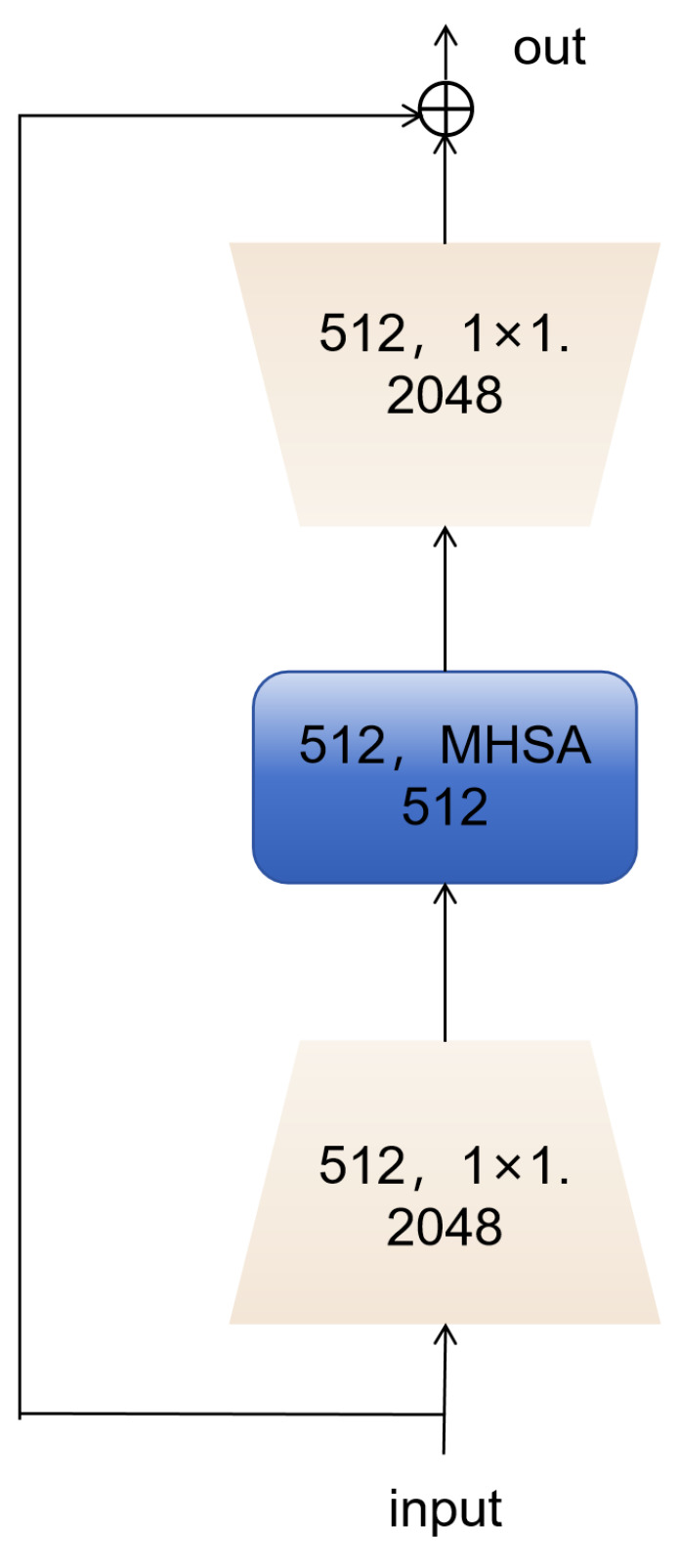
BoTNet [7].

**Figure 3 sensors-24-06911-f003:**
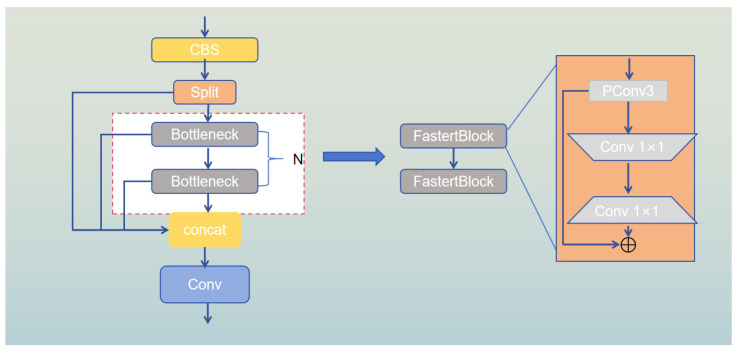
C2f_faster.

**Figure 4 sensors-24-06911-f004:**
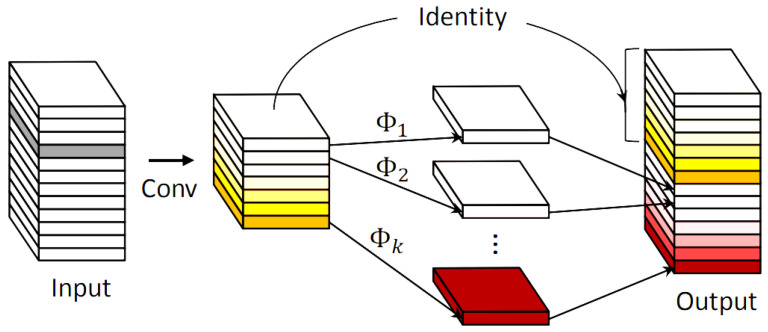
GhostConv [8].

**Figure 5 sensors-24-06911-f005:**
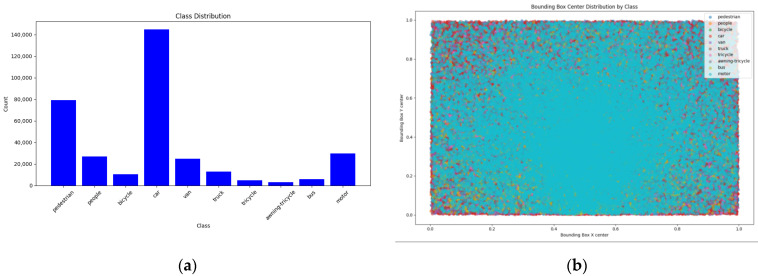
VisDone2019-DET. (**a**) Information on 10 categories in VisDrone-2019. (**b**) The location of objects in the image and the distribution of their height and width.

**Figure 6 sensors-24-06911-f006:**
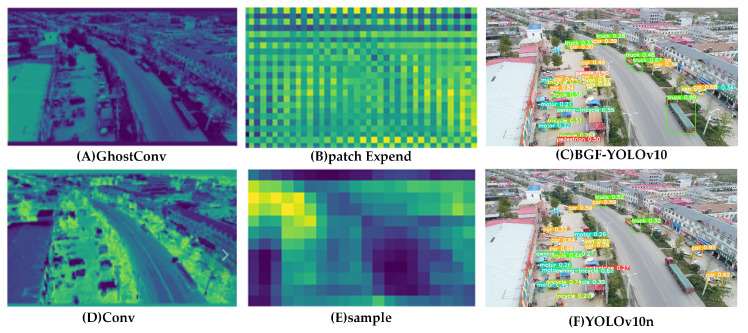
Heatmap analysis.

**Figure 7 sensors-24-06911-f007:**
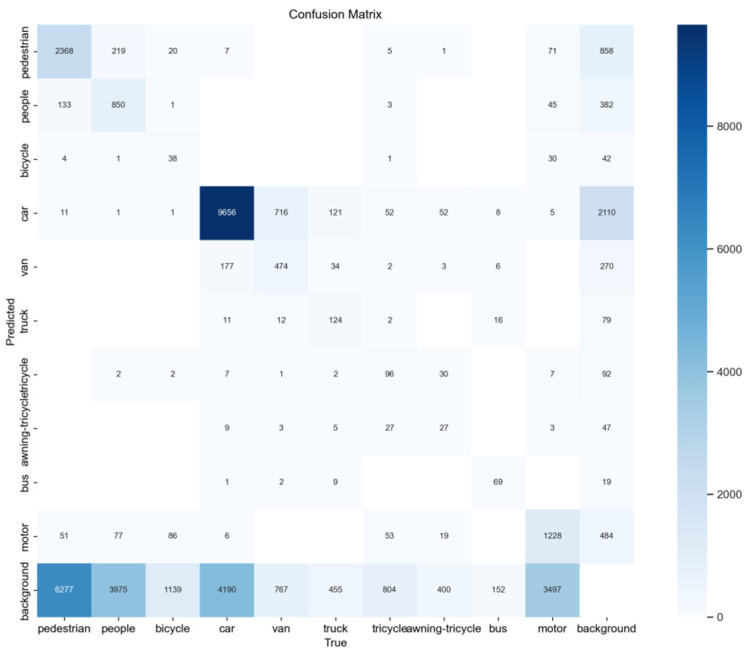
Confusion matrix analysis.

**Figure 8 sensors-24-06911-f008:**
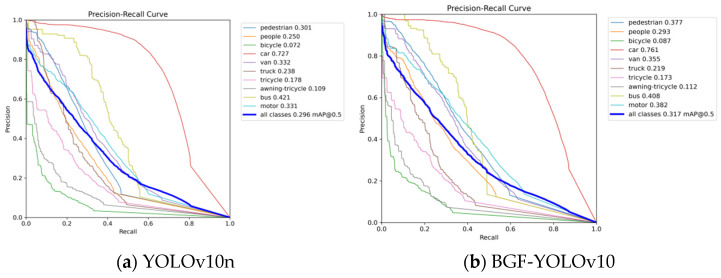
PR curve analysis.

**Table 1 sensors-24-06911-t001:** Comparison with YOLO Series.

Network	Params (M)	GFLOPs	mAP50										
			All	Pedestrian	People	Bicycle	Car	Van	Truck	Tricycle	Awing-Tricycle	Bus	Motor
YOLOv5n	1.9	** 4.2 **	0.22	0.295	0.232	0.0344	0.652	0.165	0.157	0.0964	0.0491	0.253	0.292
YOLOV8n	3.2	8.1	0.30	0.320	0.221	** 0.084 **	0.73	0.368	0.247	** 0.183 **	0.109	0.428	0.333
YOLOV10n	2.3	8.2	0.29	0.300	0.25	0.072	0.727	0.332	0.238	0.178	0.109	0.421	0.331
BGF-YOLOv10 (ours)	** 2 **	8.6	** 0.32 **	** 0.325 **	** 0.252 **	0.0813	** 0.742 **	** 0.388 **	** 0.265 **	0.12	** 0.125 **	** 0.443 **	** 0.362 **

**Table 2 sensors-24-06911-t002:** Comparison of detection results between BGF-YOLOv10 and current existing methods on the VisDrone dataset.

Network	mAP	Params (M)	FPS
Faster-RCNN [34]	25.4	41.4	27.9
Cascade-RCNN [35]	26.2	69.2	22.6
YOLOX-L [36]	31.4	54.2	18.3
CDMNet [37]	31.9	23.8	-
DCRFF [38]	35.0	-	-
YOLOv7	35.5	37.2	** 43.1 **
BGF-YOLOv10 (ours)	** 39.5 **	** 2 **	37.0

**Table 3 sensors-24-06911-t003:** Comparison of detection results between BGF-YOLOv10 and current existing methods on the UAVDT dataset.

Network	mAP50	mAP	FPS
DSHNet [39]	30.4	17.8	16.4
SODNet [40]	29.9	17.1	45.0
YOLOv5	41.3	26.9	25.1
YOLOv8	40.0	24.2	53.8
BGF-YOLOv10 (ours)	42.0	27.2	44

**Table 4 sensors-24-06911-t004:** Ablation Effects of Each Module.

Model					Parameters (M)	Precision (%)	Recall	mAP50	GFLOPs
BoTNet	Faster_block	Ghostconv	Detect Head	PatchExpand					
					2.3	0.39	0.305	0.293	8.2
√					4	0.42	0.32	0.317	12
	√				2	0.39	0.30	0.303	8.0
		√			1.9	0.393	0.306	0.29	8.2
			√		4	0.409	0.321	0.31	17
				√	2	0.393	0.306	0.298	8.5
√	√				2.6	0.41	0.32	0.31	10.8
√	√		√		4	0.413	0.316	0.312	8.7
√	√	√		√	3	0.38	0.31	0.30	7.5
√	√	√	√	√	2	0.415	0.32	0.321	8.6

## Data Availability

VisDrone is available at that address: GitHub—VisDrone/VisDrone-Dataset: The dataset for drone based detection and tracking is released, including both image/video, and annotations. UAVDT is available at that address: https://sites.google.com/view/grli-uavdt/%E9%A6%96%E9%A1%B5, accessed on 17 September 2024.

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
