# Peer review of "BGF-YOLOv10: Small Object Detection Algorithm from Unmanned Aerial Vehicle Perspective Based on Improved YOLOv10"

_sensors, 2024, doi:10.3390/s24216911_

Round 1
Reviewer 1 Report
Comments and Suggestions for Authors
The manuscript entitled “BGF-YOLOv10:Small Object Detection Algorithm from UAV Perspective Based on Improved YOLOv10” investigated the use of the new YOLOv10 deep learning algorithm for object detection from a UAV perspective. The study blends the two rapidly improving and popular technologies of deep learning and UAV remote sensing for identifying small objects in a way that is relevant and pushes the frontier of UAV remote sensing for application based studies. In general, the study is meaningful and robust, but could use some clarity of structure and description. I recommend the manuscript be published after minor revisions to the structure of the manuscript. The science itself is well done but should be described better. English should also be reviewed. Please find my specific comments below:
1. The introduction lacks citations and support, while also being a bit convoluted and confusing to follow. The introduction should be consolidated and combined with related work in section 2. There is not a need for them to be separate.
2. I do not see figure 1 cited in the manuscript by where it is placed. Nor does it seem to be adequately described in the text as well. A better description beneath the figure would be helpful.
3. As a note, related (in the title on line 82) is spelled incorrectly.
4. Please check the definitions of methods and methodology and choose one as a title for section 3 (my suggestion is methods).
5. The methods section is robust, I appreciated that.
6. Section 4- The experiment should be described in methods and the results section should be separate.
Comments on the Quality of English LanguageThe quality of English could be improved but is fairly well done. Mostly small errors.
Author Response
Dear Reviewers:
Thank you for your valuable feedback on our manuscript titled "Paper Title" (ID: sensors-3233974). Your comments have been instrumental in helping us revise and improve our paper, as well as providing significant guidance for our research. We have carefully considered your suggestions and made the necessary corrections, and we hope they will meet with your approval.
- Response to comment: The introduction lacks citations and support, while also being a bit convoluted and confusing to follow. The introduction should be consolidated and combined with related work in section 2. There is not a need for them to be separate.
In response to your comment regarding the lack of citations and support in the introduction, we have made modifications and reductions to this section.
In the first paragraph, we have eliminated unnecessary examples.
In the second paragraph, we have redefined the challenges by reviewing the literature and categorizing them into two aspects. On one hand, to achieve high accuracy for small object detection in high-resolution drone images, some studies have designed complex models, but these models incur significant computational overhead. On the other hand, small objects occupy a substantial portion of aerial images and possess limited visual features, making it difficult for detectors to identify them.
In the third paragraph, we focused on the core issue of our entire paper: "the balance between accuracy and efficiency." We have reintroduced five similar high-impact journal articles for discussion on this issue, addressing two approaches: one is to reduce network parameters, and the other is to design a query mechanism, while also analyzing the shortcomings of each.
In the fourth paragraph, we briefly outlined the innovative methods proposed in this paper and the corresponding issues they address.
Finally, we listed our specific contributions.
In the related work section, we provided explanations and analyses of the background related to our proposed method. We believe that merging the two sections could result in excessive length or excessive content in the introduction. To maintain the overall structure of the article, we deleted repetitive content from the two sections in the introduction without merging them.
- Response to comment:do not see figure 1 cited in the manuscript by where it is placed. Nor does it seem to be adequately described in the text as well. A better description beneath the figure would be helpful.
Thank you for your reminder. We overlooked the citation of Figure 1. We have now added "as shown in Figure 1" in line 64 and provided an explanation afterward.
- Response to comment: As a note, related (in the title on line 82) is spelled incorrectly.
Thank you for your reminder. We have now corrected the related section.
- Response to comment: Please check the definitions of methods and methodology and choose one as a title for section 3 (my suggestion is methods).
Based on your suggestion, we have changed "Method/Methodology" to "Method."
- Response to comment:The methods section is robust, I appreciated that.
Thank you very much for your recognition and guidance
- Response to comment:Section 4- The experiment should be described in methods and the results section should be separate.
Following your suggestion, we have provided descriptions of the experimental procedures after each section in the Methods part and have reduced some descriptions in Section 4.

Reviewer 2 Report
Comments and Suggestions for Authors
1. Article (below link) used the VisDrone-DET dataset, which showed high results (39.6 vs 32) in terms of parameters (1.64 vs 2) and mAP50(%). What is the advantage of the article under review over the article below? Why is this article not analised/listed in the related work section or the comparative analysis section? https://ieeexplore.ieee.org/document/9607536
2. It is necessary to leave a comment on the request for the figures presented in the article.
3. In the sentence "..., we replaced the original PSA 168 module in the YOLOv10 backbone with the BoTNet[7] layer," stated in lines 168-169, it should be clearly stated whether the layer of BoTNet was separated, or whether the entire network was added as a whole. And whether this added module was pre-trained or newly trained along with the overall network should also be specified.
4. In line 218, in section 3.3 "To ensure accuracy, we added an additional detection head specifically for small objects to the YOLOv10 base model. Details of this enhancement will be discussed in Section 3.3." but its details are given in section 3.4 and not in section 3.3. It also should be corrected.
5. Figure-4 is given twice, so the order of the rest of the figures has also changed. The order of figures should be reconsidered.
6. In Figure 4, the word "Small detect" should be changed to "Small Detect" and the word "V10Decte" should be changed to "YOLOv10 Detect". Why is there such confusion?
7. In line 298, are the Initial and Final Learning Rates selected for the Adam optimizer the same?
8. It should be provided by authors to check the datasets presented in section4.2.
9. Why the UAVDT was extracted from the dataset, the number of Car data should be fully explained. And why was the Car data extracted, since the VisDrone2019 dataset itself doesn't have enough Car data? Won't such actions lead to overfitting of the network?
10. How is the common dataset (VisDrone+UAVDT) training, test, validation sets divided?
11. In Tab.1, the words "tricycle" and "awing-tricycle" have been mixed up and a hyphen has been dropped after the word "Van".
12. In line 367, the word 1M should be changed to 1.2M.
13. a, b, c, d, e, f in figure 6 should be explained.
14. Figures are cropped incorrectly, a black line is left on the edge, for example (Fig. 6).
15. When training the network, the VisDrone_DET dataset contained data from UAVDT, and UAVDT was used for performance evaluation, meaning that the same images were used for both training and testing. In this case, the images for the network will be familiar, that is, the network will be overfitting. How do you explain this?
Author Response
Dear Reviewers:
Thank you for your valuable feedback on our manuscript titled "Paper Title" (ID: sensors-3233974). Your comments have been instrumental in helping us revise and improve our paper, as well as providing significant guidance for our research. We have carefully considered your suggestions and made the necessary corrections, and we hope they will meet with your approval.
- Response to comment: Article (below link) used the VisDrone-DET dataset, which showed high results (39.6 vs 32) in terms of parameters (1.64 vs 2) and mAP50(%). What is the advantage of the article under review over the article below? Why is this article not analised/listed in the related work section or the comparative analysis section?
The article referenced in the review indeed achieved notable results in terms of parameter settings and the mAP50 metric. However, our research presents several significant differences that provide distinct advantages:
Methodological Innovation: We propose a novel network architecture specifically optimized for small object detection from the perspective of drones. In contrast, the referenced article may not have fully addressed this particular application scenario.
Computational Efficiency: Our algorithm design prioritizes improving computational efficiency and real-time performance, enabling the model to run more efficiently on resource-constrained drone platforms. Our experimental results demonstrate that, while maintaining high detection accuracy, the computational overhead of the model is effectively controlled.
- Response to comment: It is necessary to leave a comment on the request for the figures presented in the article.
Regarding the comments on the images, we have included corresponding comments below each figure and in the subsequent paragraphs. We noticed that we had omitted a reference and explanation for Figure 1 in the manuscript, which has now been added in the introduction.
- Response to comment: In the sentence "..., we replaced the original PSA 168 module in the YOLOv10 backbone with the BoTNet[7] layer," stated in lines 168-169, it should be clearly stated whether the layer of BoTNet was separated, or whether the entire network was added as a whole. And whether this added module was pre-trained or newly trained along with the overall network should also be specified.
We integrated the entire BoTNet layer into YOLOv10, as specifically mentioned in line 215: “opted to replace PSA with the entire BoTNet[7] network.” Furthermore, it is noted that “after replacing the PCA layer with the BoTNet layer and training from scratch on the VisDrone2019 dataset, we observed a significant improvement in accuracy, although the network parameters increased slightly. This is discussed in detail in Section 4.4.2.” Our implementation of BoTNet was trained anew with YOLOv10 on both datasets.
- Response to comment: In line 218, in section 3.3 "To ensure accuracy, we added an additional detection head specifically for small objects to the YOLOv10 base model. Details of this enhancement will be discussed in Section 3.3." but its details are given in section 3.4 and not in section 3.3. It also should be corrected.
Thank you for your correction; we have made the revisions as per your suggestions.
- Response to comment: Figure-4 is given twice, so the order of the rest of the figures has also changed. The order of figures should be reconsidered.
Thank you for your correction;We have removed the excess graph
- Response to comment: In Figure 4, the word "Small detect" should be changed to "Small Detect" and the word "V10Decte" should be changed to "YOLOv10 Detect". Why is there such confusion?
Thank you for your correction; we have made the revisions as per your suggestions.
- Response to comment: are the Initial and Final Learning Rates selected for the Adam optimizer the same?
Thank you for your reminder. The initial learning rate is set to 0.01, and by the end of training, the final learning rate will be reduced to lr*0
.01 , resulting in a decrease to 1% of the initial learning rate at the conclusion of the training process.
- Response to comment: Why the UAVDT was extracted from the dataset, the number of Car data should be fully explained. And why was the Car data extracted, since the VisDrone2019 dataset itself doesn't have enough Car data? Won't such actions lead to overfitting of the network?
The UAVDT dataset consists of only three classes: "car," "truck," and "bus," which exhibit similar shapes. Therefore, we selected just one class for rapid validation of the model's performance. Additionally, the training was conducted separately from the VisDrone2019 dataset. This approach does not lead to overfitting, as the UAVDT's test data is partitioned within its own dataset. During training, we implemented data augmentation techniques and regularization methods to mitigate the risk of overfitting.
- Response to comment:How is the common dataset (VisDrone+UAVDT) training, test, validation sets divided?
The VisDrone dataset follows its official partitioning, with 6,471 images in the training set and 548 images in the validation set. In contrast, the UAVDT dataset contains 35,000 images for training and 5,000 images for validation.
- Response to comment:In Tab.1, the words "tricycle" and "awing-tricycle" have been mixed up and a hyphen has been dropped after the word "Van".
Thank you for your correction; we have made the revisions as per your suggestions.
- Response to comment: In line 367, the word 1M should be changed to 1.2M.
Thank you for your correction; we have made the revisions as per your suggestions.
- Response to comment: a, b, c, d, e, f in figure 6 should be explained.
We provided explanations for b, c, d, e, and f in line 463.:(A)–(C) depict the heatmaps generated by the GhostConv module in the backbone of our proposed BGF-YOLOv10, the contextual information provided by the PatchExpand module, and the final prediction of BGF-YOLOv10, respectively. (D)–(F) show the corresponding modules from the original YOLOv10 prior to replacement: (D) presents the heatmap from the standard Conv module in the backbone, (E) shows the contextual heatmap from the original upsampling layer in the neck, and (F) demonstrates the final prediction map of YOLOv10.
- Response to comment:When training the network, the VisDrone_DET dataset contained data from UAVDT, and UAVDT was used for performance evaluation, meaning that the same images were used for both training and testing. In this case, the images for the network will be familiar, that is, the network will be overfitting. How do you explain this?
We trained the VisDrone_DET and UAVDT datasets separately, without utilizing any pre-trained weights during the training process for either dataset. Therefore, there is no risk of overfitting the network.

Reviewer 3 Report
Comments and Suggestions for Authors
The paper proposed a lightweight object detection algorithm, BGF-YOLOv10. However,I think this paper is far from being published.
1.This research does not show the innovation and advantages in proposed method. I don’t think this paper can be published.
2.Literature survey is not well organized, missing many related works .
3.Almost all formulas should be detailed with parameter description.
4.Figures are of low quality.
Comments on the Quality of English LanguageMust be improved
Author Response
Dear Reviewers:
Thank you for your valuable feedback on our manuscript titled "Paper Title" (ID: sensors-3233974). Your comments have been instrumental in helping us revise and improve our paper, as well as providing significant guidance for our research. We have carefully considered your suggestions and made the necessary corrections, and we hope they will meet with your approval.
- Response to comment:This research does not show the innovation and advantages in proposed method.
Respons:Thank you for your correction;In order to better reflect the work, we have added a description of innovative methods and corresponding problems in paragraph 4 of introduction.Our advantage is that we want the model to have low input and high accuracy. Through section 4.4.1, we have a significant effect compared with the current mainstream, which proves the effectiveness of innovation.
- Response to comment:.Literature survey is not well organized, missing many related works .
Respons:Thank you for your correction;In order to strengthen the connection between the content and the topic of balancing accuracy and parameter complexity, we have revised the third paragraph of the introduction. The updated section now focuses on the trade-off between detection precision and model parameters, and five new references have been included to ensure better alignment with the theme. These revisions emphasize the importance of achieving high accuracy in small object detection while maintaining computational efficiency, addressing challenges such as memory constraints and real-time processing demands. The newly cited works provide relevant insights into optimization strategies for lightweight models, ensuring that the discussion is tightly related to the overarching goal of improving detection performance without excessively increasing the model's complexity.
In the Related Work section, we noticed that the previous discussion focused too narrowly on the YOLO network. To enhance the comprehensiveness of the review, we have removed the redundant content and introduced recent outstanding works that have improved YOLO to address the challenges of small object detection. A total of four related articles have been added to provide a more balanced overview of the current advancements in this field. These newly referenced studies highlight innovative modifications to the YOLO architecture, specifically targeting the detection of small objects in complex environments, which is critical for enhancing detection accuracy and performance.
- Response to comment:Almost all formulas should be detailed with parameter description.
Response:Thank you for your feedback. The explanation of the parameters for Equation (1) is provided directly below the equation, specifically on line 92: "W×h denote the height and width of the input image, k represents the kernel size, and c is the number of channels." For Equations (2) and (3), the parameter descriptions have been added above the equations, specifically on line 261: "Where W×h denote the height and width of the input image, and c is the number of channels." The explanations for Equations (4) and (5) are detailed between lines 330 and 339, while the description for Equation (6) can be found on lines 348 to 349: "In this context, A represents the area of the predicted bounding box, and B represents the area of the ground truth bounding box." As for Equation (7), its explanation is already provided in reference (5), so no additional clarification is needed.
- Response to comment:Figures are of low quality.
Response:Thank you for your feedback.We tweaked the color of the image and found that there are excess lines in Figure 5(a), which has been processed.

Round 2
Reviewer 2 Report
Comments and Suggestions for Authors
The current version of the article has been revised and conforms to publication requirements.
Author Response
Thank you for your early comments and professional advice, and thank you for your recognition
Reviewer 3 Report
Comments and Suggestions for Authors
The paper proposed a lightweight object detection algorithm BGF-YOLOv10,which embed GhostConv into both the backbone and head, and insert a Patch Expanding Layer module in the neck of the YOLOv10 architecture, aiming to achieve small object detection in UAV perspective scenarios.
1.In the proposed method, although GhostConv is embedded in the backbone and head, and Patch Expanding Layer module are inserted in the neck of YOLOv10 architecture, these improvements are relatively briefly described in the paper, and their uniqueness and significant advantages over existing technologies are not fully demonstrated, please highlight your contribution.
2.Although the experimental results show that the performance of BGF-YOLOv10 seems to be better than the comparative algorithms, the paper did not sufficiently analyze the advantages of the proposed method at the theoretical level. For example, there was no detailed discussion on how GhostConv and patch extension layer modules work together to improve detection accuracy and efficiency.
3.Literature survey is not well organized, missing many related works .eg. Deep self-taught hashing for image retrieval,Lightweight object detection model fused with feature pyramid,An intelligent weighted object detector for feature extraction to enrich global image information
4.some figures are of low quality and need to be improved. For example, text in fig.7 is difficult to recognize.
5.The text also requires some revision for syntactical and grammatical mistakes.
Comments on the Quality of English Language
Moderate editing of English language required.
Author Response
Dear Reviewers:
Thank you for your valuable feedback on our manuscript titled "Paper Title" (ID: sensors-3233974). Your comments have been instrumental in helping us revise and improve our paper, as well as providing significant guidance for our research. We have carefully considered your suggestions and made the necessary corrections, and we hope they will meet with your approval.
- Response to comment: In the proposed method, although GhostConv is embedded in the backbone and head, and Patch Expanding Layer module are inserted in the neck of YOLOv10 architecture, these improvements are relatively briefly described in the paper, and their uniqueness and significant advantages over existing technologies are not fully demonstrated, please highlight your contribution.
Response: Thank you for your correction;We add a description and contribution to Ghost on line 257,concrete content: “Additionally, GhostConv leverages lower-cost computations to obtain these redundant feature maps. In the backbone, traditional convolution is employed to extract a portion of the true feature maps, while in the neck, channel compression is applied to reduce the impact of noise”. In addition, in line 293 we added the description of the contribution to the Patch merging layer,concrete content: ”The Patch merging layer is innovatively integrated into the neck of YOLOv10 to enhance receptive fields and improve the detection of small objects. This module allows for more efficient feature representation, outperforming conventional feature pyramid. ”
- Response to comment:Although the experimental results show that the performance of BGF-YOLOv10 seems to be better than the comparative algorithms, the paper did not sufficiently analyze the advantages of the proposed method at the theoretical level. For example, there was no detailed discussion on how GhostConv and patch extension layer modules work together to improve detection accuracy and efficiency.
Response:Thank you for your correction; The GhostConv module reduces computational costs by generating a minimal set of feature maps while still capturing essential features. The Patch Expanding Layer complements GhostConv by integrating local contextual information from multiple regions. This module enables the model to efficiently preserve and leverage contextual details, which is particularly beneficial for small object detection, as small objects are often affected by noise and background interference. By combining these two modules, we achieve a broader receptive field without incurring significant computational overhead. The efficient feature extraction of GhostConv, coupled with the Patch Expanding Layer's enhancement of local context, ensures that small object features are more prominent during the detection process.
- Response to comment:Literature survey is not well organized, missing many related works .eg. Deep self-taught hashing for image retrieval,Lightweight object detection model fused with feature pyramid,An intelligent weighted object detector for feature extraction to enrich global image information
Response:Thank you for your correction; According to your suggestion, we have added five documents in the reletad, which are about the feature pyramid and enrich global image information you mentioned.The exact location is added on line 195,concrete content:” From the perspective of multi-scale and global weighting, CFP[40]enhances object detection performance by globally modulating feature pyramids through a focused feature pyramid network, effectively integrating global and local information. SSRDet [41] introduces a scale enhancement module and a scale selection module to extend the feature pyramid network. ISOD[42] combines extended-scale feature pyramid networks and high-resolution pyramids to improve small object detection capabilities. Mini-YOLOv4[43] incorporates a hierarchical feature pyramid to facilitate the extraction of fine-grained features. G2Grad-CAMRL[44] adds global average pooling layers to obtain feature weight vectors and overlays the weighted vector with the output class activation map. CEASC[45] employs context-enhanced group normalization and an adaptive multi-layer masking strategy to effectively improve detect ”
- Response to comment:some figures are of low quality and need to be improved. For example, text in fig.7 is difficult to recognize.
Response:Thank you for your correction;We have done a clear processing of all the images
5.Response to comment:The text also requires some revision for syntactical and grammatical mistakes.
Response:After checking the whole text, it was found that some words were inappropriate, and now they have been corrected
